# How and why psychologists should respond to the harms associated with generative AI

Laura G. E. Smith, Richard Owen, Alicia Cork & Olivia Brown

Innovations in generative AI to create human-like text, images, and videos can cause social, psychological, and political harms. Here, we explain how psychologists can mobilize their extensive theoretical and empirical resources to better anticipate, understand, and mitigate those harms.

## A matter of influence

Miquela Sousa has 2.6 million followers on Instagram. Her feed looks like that of a typical 19-year-old woman – but she is not human. She is a virtual influencer: a high-fidelity, hyper-realistic metahuman, comprised of synthetic media generated by artificial intelligence. The generative artificial intelligence (or Gen-AI) used to create such media is underpinned by machine learning (ML) models trained on billions of data points to generate multi modal, visual and/or auditory synthetic media in response to user prompts. The generated media can be microtargeted: personalized and tailored to any audience, to have maximum influence. This has benefits, for example, Gen-AI can potentially improve the efficiency of creative work. However, due to their potential for influence, outputs of Gen-AI like Miquela are, and will continue to be, socially, psychologically, and politically disruptive, and ethically entangled.

Whilst the ethical entanglements of Gen-AI are a topic of both interest and active research in computer and social science, we argue that research questions that are fundamental to the aims and scope of psychology are, to date, conspicuous by their absence. Yet, responsible innovations in this technology require development of a systematic research agenda in Psychology, which should aim to facilitate a step change in our understanding of the role of people – and our data – in the socio-technical systems in which we interact with synthetic media and Gen-AI technologies. Indeed, Gen-AI and its outputs raise research questions that are directly relevant to Psychologists, for example: how could psychological research be (mis)used to enhance the influence of Gen-AI outputs? How and why could the outputs of Gen-AI affect people? How can we use knowledge of these processes to help ensure Gen-AI and synthetic media are developed and used responsibly?

The topic of social influence is, after all, at the very heart of the discipline of social psychology. Therefore, we argue that psychologists need to urgently engage (more) with research on Gen-AI as both a nascent field of enquiry within psychology, and as an interdisciplinary programme aimed at minimizing harms and fostering responsible innovation[1].

## Concerns for psychologists

Psychologists should be concerned with Gen-AI technology because of the way in which it could leverage people's data and insights from psychological research to have maximum influence on its target audience. Every design decision about the affordances of Gen-AI and its outputs has both predictors and implications, therefore studies of the decisions of developers and users as well as how they may impact other people are critical to ensuring responsible innovation. In some ways, this is true of technologies of influence that go back many decades. But, in combination with other technologies such as recommendation algorithms and bots, what is of both value and concern is the scale by which Gen-AI has to the potential to co-opt psychological processes and phenomena.

Of primary concern is the ability for Gen-AI to create hyper-realistic, relatable synthetic media that can be used to spread (mis/dis)information[2] at scale, and to microtarget individuals. By using personality inference technology[3], hostile actors can cause polarization of large numbers of people, interfere in democratic political processes[4], and even incite violence and hatred or motivate terrorism[5] – creating significant societal disruption. But it is not only these nefarious uses that should be of concern to psychologists: it is also the legitimate use of such hyper realistic synthetic media, from business and marketing to education and entertainment.

The role that psychologists have to play in understanding the design and impacts of Gen-AI cannot be underestimated[1]. Gen-AI is embedded in a sociotechnical system – made by humans, trained on humans' data, and with a human audience. Psychologists can provide important insights and explanatory frameworks for such sociotechnical systems, but at the same time we need to be responsible about how we conduct such research and the insights we disclose. The move towards open science means that our insights published in open access papers could be available as training data for large language models (LLMs). That is, Gen-AI tools like ChatGPT can synthesize the outputs of psychological research and use this to suggest more advanced ways to design synthetic media to manipulate and influence a target audience. Insights from Psychological research are known to have been used to influence and manipulate people: for example, data on personality and social media behavior were used by Cambridge Analytica to influence election outcomes[6]. For this reason, journals such as *Communications Psychology* ask authors to consider the dual use implications of their research. Now, because open science practices mean that more insights about psychological processes are openly available, and because the skills barrier for using Gen-AI is becoming ever lower, using Gen-AI technologies to leverage psychological processes could no longer be the domain of a few gatekeepers. The implication is that now (notwithstanding recently implemented guardrails) anyone with access to Gen-AI tools like ChatGPT and GPT-4o can write (almost) any code[7], or any content. The flip side is that the very psychological models that could be used to design and increase the influence of synthetic media could also provide explanatory frameworks for the harms that result, and how to anticipate and mitigate them, and engage in more responsible innovation.

In recognition of the gravity of the harms that may come from unregulated use of such technology, world leaders and representatives of big tech firms met in November 2023 for the AI Safety Summit. There are some promising initiatives, including regulation in the EU (the AI Act of the EU)

and Brazil, a blueprint for an AI Bill of Rights in the US, the development of responsible practices for synthetic media and a call for collective action to develop responsible practices. However, psychologists have been conspicuously absent from international discussions and debates such as the Responsible AI initiative and the AI Safety Summit fringe event, which were largely policy or technologist-led. Similarly, guidance on responsible use of Gen-AI from the private sector (e.g., Deloitte, PwC) insufficiently draws upon psychological research and insights, even if some companies are taking a proactive approach to the ethical creation and use of Gen-AI avatars, with policies and processes based on principles of consent, control, and collaboration (e.g., www.synthesia.io/ethics; https://syntheticmedia.partnershiponai.org).

Why have Psychologists not been invited to the table, and how can we get a seat? One reason may be that programmatic work in Psychology on Gen-AI has not been funded yet. We conducted an analysis of research projects funded in 2022-2023 by major research councils in the US (National Science Foundation), Canada (Social Sciences and Humanities Research Council), the UK (UK Research and Innovation), Europe (Horizon), and Australia (Australian Research Council). In total, 67,574 projects were funded, and 3.23% (2182) of these projects were relevant to AI. Of these, only 3 (0.14%) were relevant to Gen-AI and included a psychology component. None are led by Psychology.

## A research agenda
We suggest three priority research areas that urgently require Psychologist-led research programmes. The first involves investigating how the affordances and features of Gen-AI technologies harness and change psychological processes – so that potential harms can be understood, anticipated, and mitigated. Whilst awareness of the risks of Gen-AI is increasing, it remains the case that our current theories cannot easily be applied to explain the impact of the affordances of these new technologies[8]. Therefore, a psychological research agenda should prioritize research that aims to understand how the features and affordances of Gen-AI technology interact with social and psychological variables. For example, research could investigate the factors that affect people's vulnerability and resilience to being influenced by AI-generated media, through investigating how psychological variables moderate the impact of the affordances of specific technical features on (for example) the perceived trustworthiness, persuasiveness, humanness, and relatability of such media – and outcomes such as attitude, norm, and behavior change. Then, psychologists should investigate and propose mitigations "by-design" that can reduce contextual vulnerability and increase resilience to harms within these sociotechnical systems. A related question that future research should target is how Gen-AI could be prevented from accessing these insights and training itself to improve its performance.

The second priority research area involves using the output of Gen-AI to investigate the biases that exist within societies (some of which may be the variables described above). LLMs learn patterns of bias that exist within training data, and, in turn, outputs of those models reinforce those biases[9]. These systems offer a novel opportunity for psychologists to study and compare the biases and implicit knowledge inherent within training datasets. We can view the training data not as providing unbiased ground truth, but instead as social phenomena worth examination; Gen-AI can serve as a diagnostic tool to identify, understand, and measure biases. Related research questions include how the nature of the training data impact on the behavior of the target audience through affecting Gen-AI outputs, and how data from such behavior feed forward into these outputs as training data. By employing experimental methodologies that go beyond the mathematical approaches commonly employed by technical experts, psychologists are well-placed to provide nuance to the conversation around biased AI.

Third, to better anticipate (and prevent) harms by design, psychologists need to collaborate with technologists and academics from other disciplines using mechanisms such as the responsible research innovation framework of UKRI. This would generate questions that would not otherwise be asked, leading to a better understanding of the assumptions baked into Gen-AI and the underlying ML that would not otherwise be recognized. This interdisciplinary approach has already been used, for example, to identify algorithmic contingencies that can affect future outcomes, uncertainties, and harms of ML models[10]. In the same way, psychologists can help identify contingencies that could affect the potential of Gen-AI technologies to cause – or prevent – harm. In particular, the "Anticipate - Reflect - Engage - Act" (AREA) model[1] can help psychologists to generate research questions and elucidate the factors that can give rise to contextual vulnerability.

## A hopeful outlook
Psychologists have an opportunity to develop and test theories that anticipate when and why personal, social, and political disruption can be produced through affordances of Gen-AI technology. Using these insights, and by working with technologists and academics in other disciplines, mitigations could be baked into Gen-AI technologies by design. Furthermore, by using the output of Gen-AI to understand biases that are embedded in training data, and how the outputs affect people, psychologists can advise on how LLMs can use more appropriate training data, and even curate or synthesize training datasets that may work with Gen-AI to reduce societal inequalities. This means that psychologists can help others to use Gen-AI for the betterment of - rather than to the detriment of - society. Through research in these areas, raising our collective voice, and engaging responsibly with government, regulators, industry, and other disciplines, Psychologists can help ensure a future in which Gen-AI works to support - rather than harm – people and society.

**Laura G. E. Smith** [1] ✉, **Richard Owen**[2], **Alicia Cork** [1] & **Olivia Brown** [1]
¹University of Bath, Bath, UK. ²University of Bristol, Bristol, UK.
✉e-mail: l.g.e.smith@bath.ac.uk

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

### Author contributions

This statement of author contributions was prepared using the CRediT (Contributor Roles Taxonomy) guidelines. Laura Smith contributed to: Conceptualization, Writing-Original Draft, Writing-Review and editing. Richard Owen contributed to: Conceptualization, Writing-Original Draft, Writing-review and editing. Alicia Cork contributed to: Writing-Original Draft, Writing-review and editing. Olivia Brown contributed to: Writing-Original Draft, Writing-review and editing.

### Competing interests

The authors declare no competing interests.
