## [Peer Review File · Communications Psychology]

27th Feb 24

Dear Laura,

Thank you for your patience during the peer-review process. Your manuscript titled "Why Psychologists Should be (More) Concerned with the Harms Associated with Generative-AI" has now been seen by 2 reviewers, and I include their comments at the end of this message.

The reviewers find some merit in the Comment, and highlight that it does pertain to a rapidly evolving topic. At the same time, they criticise the lack of novelty or new degree of depth in the proposal (especially for a general audience), and some claims regarding the field's ignorance about the developments.

The originality of a contribution is a key criterion for a Comment in Communications Psychology; moreover, Comments should be forward-looking.

We would be willing to look at a revised version, provided it integrates the referees' feedback and achieves the following goals:

- 1) Limit overlap with existing commentary, especially in terms of reviewing general developments
- 2) Make an original contribution specifically to the field of psychology (i.e. highlight the dangers/opportunities for psychology/psychologists in greater depth)
- 3) Strengthen the section containing recommendations and implications

We would be happy to receive a revision that satisfies these issues for editorial consideration and potential further peer review. However, please be aware that we would not contact the referees again if the revision did not engage thoroughly with the referees' concerns to satisfy the three key issues listed above.

EDITORIAL POLICIES AND FORMATTING

You will find a complete list of formatting requirements following this link:

<https://www.nature.com/documents/commsj-style-formatting-checklist-review-perspective.pdf>

Please use the checklist to prepare your manuscript for resubmission.

* **TRANSPARENT PEER REVIEW:** Communications Psychology uses a transparent peer review system. This means that we publish the editorial decision letters including Reviewers' comments to the authors and the author rebuttal letters online as a supplementary peer review file. We publish these records for all accepted manuscripts. However, on author request, confidential information and data can be removed from the published reviewer reports and rebuttal letters prior to publication. If your manuscript has been previously reviewed at another journal, those Reviewers' comments would not form part of the published peer review file.

If you have any questions about any of our policies or formatting, please don't hesitate to contact me.

Please use the following link to submit your revised manuscript and a point-by-point response to the referees' comments (which should be in a separate document to any cover letter):

[link redacted]

We hope to receive your revised paper within 8 weeks; please let us know if you aren't able to submit it within this time so that we can discuss how best to proceed. If we don't hear from you, and the revision process takes significantly longer, we may close your file.

Please do not hesitate to contact me if you have any questions or would like to discuss these revisions further. We look forward to seeing the revised manuscript and thank you for the opportunity to review your work.

Best wishes,

Marike

Marike Schiffer, PhD

Chief Editor

Communications Psychology

REVIEWERS' EXPERTISE:

Reviewer #1 (generative) AI, psychology

Reviewer #2 (generative) AI, psychology

REVIEWERS' COMMENTS:

Reviewer #1 (Remarks to the Author):

Does the Comment formulate a novel, thought-provoking take? Not exactly. The comment recapitulates some of the risks created by generative AI, and argues that more psychologists should work on that topic (with some nice examples of the kind of work they could do). I might be the wrong audience here, given that I work on this topic and know many psychologists who do. That said, the comment may be novel and thought-provoking to someone who is new to this topic.

Is the argument persuasive? Yes, in the sense that it is straightforward : generative AI is indeed a revolutionary technology, and psychologists are indeed able to contribute to mitigating its potential harms in various domains, so, they should.

I am less persuaded by the claims that psychologists are "asleep at the wheel" (a phrase which, incidentally, is not quite respectful) and ignoring these research questions. Generative AI has not been around for that long, especially if one considers when it became easily available. Things started to be hot one year ago with the mass release of ChatGPT; many, many colleagues I know started research projects on generative AI in the months that followed, and I did, too. So I would

expect that it is just a matter of time before we see many publications on this topic. The fact they're not all out yet does not signal a lack of interest, but reflects the pace of the research process.

Does the Comment reflect a viewpoint that has recently or historically not received sufficient exposure? I would say that a lot of attention has already been given to the view that generative AI creates risks. What is new is the view that not enough psychologists are working on this topic, and the call for them to do so.

Will the Comment be of interest to researchers in my field, or a wider audience of readers? For researchers in my field, the comment does not formulate novel ideas (although the idea that open psychology datasets may have dual use was new to me); but it may be of interest as a citable piece when arguing for the importance of their work.

The interest for a wider audience is not easy for me to judge, as it would depend on the readers' previous exposure to debates about generative AI. But on balance, yes, I believe this may be of interest. I would still recommend that the authors tone down their claims about the current lack of psychological research on the risks of generative AI, since it may sound disparaging for all the people who *are* doing this work at the moment.

Additionally, I understand that the argument focuses on the potential harms of generative AI, but for a broader audience, it might be good to also include at least a sentence or two about its potential benefits. In the current version of the comment, the exclusive focus on harms may lead people to wonder why we even created this technology in the first place, if its potential harms are not balanced with any benefit.

Reviewer #2 (Remarks to the Author):

First of all, I acknowledge that reviewing a comment for an academic journal is not straightforward.

Compared to regular empirical and conceptual papers, its assessment tends to be more subjective. Unfortunately, I find the current commentary to lack sufficient novelty and interest to warrant publication. In my view, the present commentary does not introduce new perspectives that would justify its publication.

It is well-established that GenAI is associated with several threats that require consideration. The five reasons why psychologists should be more concerned about AI do not expand my current understanding; it is also unclear what the exact implications of this commentary are.

The literature in various domains of the social sciences has published comments or short conceptual articles on the impact of GenAI on people, society, and decision-makers. Examples that delve much deeper into important aspects of its use include, for instance, Bockting et al. (2023) or Peres et al. (2023) in a neighboring domain.

It would be important to provide more depth; such depth could offer readers new ideas and perspectives regarding the use of GenAI. Furthermore, I found that the coherence and structure of the comment could be improved (see, Sætra (2023), for an example).

On a minor note, I think the introduction concerning virtual influencers is quite lengthy and somewhat disconnected; while describing this phenomenon of virtual influencers might capture the attention of some readers, it does not appear to be insightful to many of them.

Bockting, C. L., van Dis, E. A., van Rooij, R., Zuidema, W., & Bollen, J. (2023). Living guidelines for generative AI—why scientists must oversee its use. *Nature*, 622(7984), 693-696.

Peres, R., Schreier, M., Schweidel, D., & Sorescu, A. (2023). On ChatGPT and beyond: How generative artificial intelligence may affect research, teaching, and practice. *International Journal of Research in Marketing*. 40(2), 269-275.

Sætra, H. S. (2023). Generative AI: Here to stay, but for good?. *Technology in Society*, 75, 102372.

Response to Reviewers

COMMSPSYCHOL-23-0438-T, "Why Psychologists Should be (More) Concerned with the Harms Associated with Generative-AI"

Below, we outline the reviewers' suggestions (indented, in italics) and how we have addressed them in this revision (in bold).

Reviewer #1 (Remarks to the Author):

1. *Does the Comment formulate a novel, thought-provoking take? Not exactly. The comment recapitulates some of the risks created by generative AI, and argues that more psychologists should work on that topic (with some nice examples of the kind of work they could do). I might be the wrong audience here, given that I work on this topic and know many psychologists who do. That said, the comment may be novel and thought-provoking to someone who is new to this topic.*

Is the argument persuasive? Yes, in the sense that it is straightforward : generative AI is indeed a revolutionary technology, and psychologists are indeed able to contribute to mitigating its potential harms in various domains, so, they should.

As Reviewer 1 highlights, we hope that our work may be thought-provoking for people who are new to this topic. This is our target audience. The primary aims of our Comment are to raise awareness of relevant, urgent issues for psychologists who aren't working on Gen-AI but whom have expertise relevant to the psychological processes that Gen-AI harnesses, and to provide a call to arms to research funders and journal Editors so that it is more straightforward for Psychologists to conduct research in this area.

We now clarify the novel contributions of our *Comment* and explain them in greater depth. Our novel arguments are three-fold:

(1) Our primary novel motivation for the *Comment* is to raise awareness of how psychological insights can be weaponized - and therefore why Psychologists should engage more. On pages 4-5, we explain how the risks we described in the original Table 1 are connected, for example:

"Psychologists should be concerned with Gen-AI technology because of the way in which it leverages people's data and insights from psychological research to have maximum influence on its target audience. In combination with other technologies such as recommendation algorithms and bots, it can be used to monetize and weaponize psychological processes and phenomena². For example, hostile actors can train an LLM using the content of research articles from Psychology on persuasion and trust, and with social media communications data from the target population and public figures. Then, they can use Gen-AI to create hyper-realistic, relatable user profiles and posts, and deep fakes, then multiply them with bots and botnets. By spreading apparently highly credible, trustworthy, and relatable mis/disinformation³ at scale and microtargeting individuals by using personality inference technology⁴, hostile actors can cause polarization of large numbers of people, interfere in democratic political processes⁵, and even incite violence and hatred or motivate terrorism⁶ - creating significant societal disruption. To anticipate and mitigate the harms that may arise from this technology, an understanding of how the affordances of these technologies work with human psychology and data must be at the core of any research going forward".

(2) Connected to (1), we then explain that that open psychology papers and datasets may carry dual use concerns. We have now expanded on these novel arguments (p. 5):

“Indeed, Psychology has a responsibility to anticipate, explain, and help mitigate the harms associated with the design of any technologies that can weaponize its insights⁷. Gen-AI is embedded in a complex sociotechnical system - made by humans, trained on humans’ data, and with a human audience. The move towards open science in Psychology means that our insights in open access papers are available as training data for LLMs⁸ - but Psychology has some potentially dangerous insights that carry dual-use concerns. Psychology can inform others about how to influence, persuade and manipulate people, how to engender trust, how to build relationships, etc. And whilst Export Control legislation prevents knowledge in other disciplines being used for harm (for example, chemical weapon manufacture), insights from Psychology have not been considered under such legislation. Now, because of open science practices and because the skills barrier for using Gen-AI is becoming ever lower⁹, using these technologies to leverage psychological processes is no longer the domain of a few gatekeepers. The implication is that now (notwithstanding recently implemented guardrails, which can be overcome), anyone with access to Gen-AI tools like ChatGPT can write (almost) any code¹, or any content. The flip side is that the psychological models that are weaponized provide explanatory frameworks for the harms that result.”

(3) Points (1) and (2) lead us to our novel call to arms for Psychologists to engage with research on Gen-AI – and because of the above points, we would argue that it is urgent and much needed. To support our arguments, although as Reviewer 1 quite rightly notes that we cannot cite research that is presently being conducted in Psychology on Gen-AI (because it may be in early stages and is not in the public domain), we conducted a review of all research projects that were funded by major funding bodies in the US, UK, Australia, and Europe in 2022-23 (p. 6):

“One reason may be that programmatic work in Psychology on Gen-AI has not been funded yet. We conducted an analysis of research projects funded in 2022-2023 by major research councils in the US (National Science Foundation), Canada (Social Sciences and Humanities Research Council), the UK (UK Research and Innovation), Europe (Horizon), and Australia (Australian Research Council). In total, 67,574 projects were funded, and 3.23% (2,182) of these projects were relevant to AI. Of these, only 3 (0.14%) were relevant to Gen-AI and included a psychology component. None aim to address the issues raised in this *Comment*, and none are led by Psychology, despite the potential of Gen-AI to change the face of society through harnessing psychological processes.”

This lack of funded projects in the area suggests that (a) Psychologists in these regions have not yet mobilised to do significant, funded programmatic research on the topic, and/or (b) there are insufficient funding opportunities to support this research from these funding bodies. We now suggest potential solutions on pp. 6-9 (see below).

2. *I am less persuaded by the claims that psychologists are "asleep at the wheel" (a phrase which, incidentally, is not quite respectful) and ignoring these research questions. Generative AI has not been around for that long, especially if one considers when it became easily available. Things started to be hot one year ago with the mass release of ChatGPT; many, many colleagues I know started research projects on generative AI in the months that followed, and I did, too. So I would expect that it is just a matter of time before we see many publications on this topic. The fact they're not all out yet does not signal a lack of interest, but reflects the pace of the research process.*

Absolutely: we recognize that Gen-AI is a rapidly evolving research area. We appreciate that psychologists may be conducting research on Gen-AI that is not yet in the public domain. It may be too soon for papers to have been published on Gen-AI in Psychology, therefore (as we explain above) we conducted a (non-exhaustive) review of data of funded research projects from selected research funding bodies that make their data available, to try to find new work in this area (p. 6). This provided evidence that, to date, Psychologists have not yet begun funded programmes of research into the issues we raise in the Comment (at least within the regions the selected funding agencies cover).

3. *Does the Comment reflect a viewpoint that has recently or historically not received sufficient exposure? I would say that a lot of attention has already been given to the view that generative AI creates risks. What is new is the view that not enough psychologists are working on this topic, and the call for them to do so.*

Will the Comment be of interest to researchers in my field, or a wider audience of readers? For researchers in my field, the comment does not formulate novel ideas (although the idea that open psychology datasets may have dual use was new to me); but it may be of interest as a citable piece when arguing for the importance of their work.

As R1 recognizes, our call to arms for Psychology to engage with research on AI is novel. We agree that other excellent articles have pointed to the risks of Gen-AI, and we have cited some examples. Our motivation for the *Comment* was highlight some of the most pertinent risks from a psychological research perspective, and to argue that Psychologists should play leading role on research on them, and how to mitigate them.

In this revision, we have reduced the length of our description of around the points that previous papers have already explained in depth (see our response to R2 #2). This has given us more space to explain our novel arguments. For example, we now go into greater depth on how exactly Psychology and Psychologists can play a role in researching and mitigating those risks (see section starting on p. 7, “A Gen-AI Research Agenda for Psychology”).

Furthermore, whilst previous research has explained what some of the risks and harms of Gen-AI might be, our approach is novel because we explain how psychologists can provide unique *explanations*. Thus far, risks of Gen-AI have typically (although not exclusively) been approached from other disciplines, such as computer science, the creative arts, or education. As an antidote, we explain how psychologists can provide rigorous methods for how to better understand the psychological mechanisms by which these risks can cause harm. We also point out some unique risks such as dual use and the fact that export controls do not cover psychological insights that could be weaponized (as Reviewer 1 highlights above).

For example, on p. 5 we state:

“Psychology has a responsibility to anticipate, explain, and help mitigate the harms associated with the design of any technologies that can weaponize its insights⁷. Gen-AI is embedded in a complex sociotechnical system - made by humans, trained on humans’ data, and with a human audience. The move towards open science in Psychology means that our insights in open access papers are available as training data for LLMs⁸ - but Psychology has some potentially dangerous insights that carry dual-use concerns. Psychology can inform others about how to influence, persuade and manipulate people, how to engender trust, how to build relationships, etc. And whilst Export Control legislation prevents knowledge in other disciplines being used for harm (for example, chemical weapon manufacture), insights from Psychology have not been considered under

such legislation. Now, because of open science practices and because the skills barrier for using Gen-AI is becoming ever lower⁹, using these technologies to leverage psychological processes is no longer the domain of a few gatekeepers. The implication is that now (notwithstanding recently implemented guardrails, which can be overcome), anyone with access to Gen-AI tools like ChatGPT can write (almost) any code¹, or any content. The flip side is that the psychological models that are weaponized provide explanatory frameworks for the harms that result.”

And on page 7:

“There are three priority research areas that urgently require Psychology-led research programmes. The first involves investigating how the affordances and features of Gen-AI technologies harness and change psychological processes - so that potential harms can be understood, anticipated, and mitigated. Whilst awareness of the risks of Gen-AI is increasing, it remains the case that our current theories cannot easily be applied to explain the impact of the affordances of these new technologies¹¹. Therefore, a psychological research agenda should prioritize research that aims to understand how the features and affordances of Gen-AI technology interact with social and psychological variables. For example, research could investigate the factors that affect people’s vulnerability and resilience to being influenced by AI-generated media, through investigating how psychological variables moderate the impact of the affordances of specific technical features on (for example) the perceived trustworthiness, persuasiveness, humanness, and relatability of such media – and outcomes such as attitude, norm, and behavior change. Then, psychologists should investigate and propose mitigations that can reduce contextual vulnerability and increase resilience to harms within these sociotechnical systems.”

*The interest for a wider audience is not easy for me to judge, as it would depend on the readers' previous exposure to debates about generative AI. But on balance, yes, I believe this may be of interest. I would still recommend that the authors tone down their claims about the current lack of psychological research on the risks of generative AI, since it may sound disparaging for all the people who *are* doing this work at the moment.*

Please see our responses above to Reviewer1 #1 and #2. Reviewer 1’s point above is key: whilst both reviewers are experts working in the area of Gen-AI, we wrote this paper for psychologists who have *not* had previous exposure to debates about Gen-AI. As Reviewer 1 recommends, we have toned down our claims about psychologists showing a lack of interest in this area whilst also providing evidence of the scarcity of (funded) research in this area.

4. *Additionally, I understand that the argument focuses on the potential harms of generative AI, but for a broader audience, it might be good to also include at least a sentence or two about its potential benefits. In the current version of the comment, the exclusive focus on harms may lead people to wonder why we even created this technology in the first place, if its potential harms are not balanced with any benefit.*

Thank you for this suggestion. We agree and have elaborated on the following sentences on the potential benefits of Gen-AI:

“It has benefits, for example, it can potentially automate tasks, and improve the efficiency of creative work, productivity, and decision making.” (p. 3)

“Psychologists can advise on how LLMs can use unbiased/more appropriate training data, and even curate or synthesize training datasets that may work with Gen-AI to reduce

societal inequalities - in other words, Psychologists can help others to use Gen-AI for the betterment of - rather than to the detriment of - society.” (p. 9)

Reviewer #2 (Remarks to the Author):

1. *First of all, I acknowledge that reviewing a comment for an academic journal is not straightforward.*

Compared to regular empirical and conceptual papers, its assessment tends to be more subjective. Unfortunately, I find the current commentary to lack sufficient novelty and interest to warrant publication. In my view, the present commentary does not introduce new perspectives that would justify its publication.

It is well-established that GenAI is associated with several threats that require consideration. The five reasons why psychologists should be more concerned about AI do not expand my current understanding; it is also unclear what the exact implications of this commentary are.

Please see our responses regarding novelty and our target audience to Reviewer 1’s points 1-4, above.

We have now shortened our introduction and description of the risks and harms associated with Gen-AI, and this has given us more space to explain our novel arguments and recommendations in more depth. Specifically, we explain how points 1,2, and 5 that were in Table 1 in the original submission (removed from this revision) are connected, leading to our primary novel argument, as we explain on p. 4:

“Psychologists should be concerned with Gen-AI technology because of the way in which it leverages insights from psychological research to have maximum influence on its target audience. In combination with other technologies such as recommendation algorithms and bots, it can be used to monetize and weaponize psychological processes and phenomena¹. For example, hostile actors could train a LLM with research articles on persuasion and trust, and social media communications data from the target population and public figures, and use Gen-AI to create multiple hyper-realistic, relatable user profiles and posts, and deep fakes, then multiply them with bots and bot-nets. By spreading this content at scale and microtargeting individuals by using personality inference technology, hostile actors could cause polarization of large numbers of people, interfere in elections, and even incite violence and hatred - creating significant societal disruption². Therefore, to anticipate and mitigate the harms that may arise from this technology, an understanding of how the affordances of these technologies work with human psychology and data must be at the core of any research going forward. Indeed, Psychology has a responsibility to anticipate, understand, and help mitigate the harms associated with the design of any technologies that can weaponize its insights³.”

We also now more clearly state the implications of our arguments for a psychological research agenda:

“In these ways, by working with technologists and academics in other disciplines, psychologists can help anticipate when and explain why social and political disruption can be produced through the weaponization of Gen-AI technology. Then, by developing theories that explain how psychological processes change or are amplified by Gen-AI, mitigations could be baked into Gen-AI technologies by design. Finally, by using the output of Gen-AI to understand biases that are embedded in training data, Psychologists can

advise on how LLMs can use more appropriate training data, and even curate or synthesize training datasets that may work with Gen-AI to reduce societal inequalities - in other words, Psychologists can help others to use Gen-AI for the betterment of - rather than to the detriment of - society. Through research in these areas, and by raising our collective voice and engaging responsibly with government, regulators, industry, and other disciplines, Psychology can ensure a future in which Gen-AI works to support - rather than harm – people, organizations, and society.” (pp. 8-9)

2. *The literature in various domains of the social sciences has published comments or short conceptual articles on the impact of GenAI on people, society, and decision-makers. Examples that delve much deeper into important aspects of its use include, for instance, Bockting et al. (2023) or Peres et al. (2023) in a neighboring domain.*

It would be important to provide more depth; such depth could offer readers new ideas and perspectives regarding the use of GenAI. Furthermore, I found that the coherence and structure of the comment could be improved (see, Sætra (2023), for an example).

*Bockting, C. L., van Dis, E. A., van Rooij, R., Zuidema, W., & Bollen, J. (2023). Living guidelines for generative AI—why scientists must oversee its use. *Nature*, 622(7984), 693-696.*

*Peres, R., Schreier, M., Schweidel, D., & Sorescu, A. (2023). On ChatGPT and beyond: How generative artificial intelligence may affect research, teaching, and practice. *International Journal of Research in Marketing*. 40(2), 269-275.*

*Sætra, H. S. (2023). Generative AI: Here to stay, but for good?. *Technology in Society*, 75, 102372.*

We agree that these papers are excellent examples of how social scientists can contribute to the conversations on Gen-AI - indeed, we cite Sætra on p. 4 of our paper. We would be enthusiastic to elaborate on our arguments and provide greater depth. However, unfortunately we cannot use these examples as models for our own paper because the papers above are significantly longer than the *Comment* format at *Communications Psychology* allows (3,161 words, 4,509 words, and 3,026 words respectively). The word limit for a *Comment* in *Communications Psychology* is 1,500, which forecloses our ability to follow the models of the articles above.

Given Reviewer 2’s recommendation, we sought advice from the Editor on whether we could extend our word count in order to provide similar depth to the above articles. The advice was that we could use up to 1,800 words but no longer. Therefore, instead of lengthening the paper, to add as much depth as possible within this tight word limit, we have shortened the introduction (see Reviewer 2 #3) and reduced duplication of content around harms (see Reviewer 1 #3). This has allowed us to add depth to our arguments on pp. 6-9.

We also note that although the above papers provide useful social science perspectives, none of them are grounded in Psychology or are written by Psychologists, underlining the novelty of our psychological arguments and our call to arms for Psychology to do more.

3. *On a minor note, I think the introduction concerning virtual influencers is quite lengthy and somewhat disconnected; while describing this phenomenon of virtual influencers might capture the attention of some readers, it does not appear to be insightful to many of them.*

Please see our response to R2 #2. We have shortened the introduction and reduced the detail about virtual influencers. We also now clarify how these examples are relevant to our arguments:

“Psychologists should be concerned with Gen-AI technology because of the way in which it leverages insights from psychological research to have maximum influence on its target audience. In combination with other technologies such as recommendation algorithms and bots, it can be used to monetize and weaponize psychological processes and phenomena¹. For example, hostile actors could train a LLM with research articles on persuasion and trust, and social media communications data from the target population and public figures, and use Gen-AI to create multiple hyper-realistic, relatable user profiles and posts, and deep fakes, then multiply them with bots and bot-nets. By spreading this content at scale and microtargeting individuals by using personality inference technology, hostile actors could cause polarization of large numbers of people, interfere in elections, and even incite violence and hatred - creating significant societal disruption. Therefore, to anticipate and mitigate the harms that may arise from this technology, an understanding of how the affordances of these technologies work with human psychology and data must be at the core of any research going forward. Indeed, Psychology has a responsibility to anticipate, understand, and help mitigate the harms associated with the design of any technologies that can weaponize its insights.” (p. 4)

We have retained some of the introduction on virtual influencers because our intended audience is Psychologists who are new to Gen-AI and we felt that this content is necessary to provide the context for our later arguments.

We extend our thanks to the reviewers for their time, consideration, and insightful comments on our manuscript.

10th May 24

Dear Laura,

Your Comment titled "Why Psychologists Should be (More) Concerned with the Harms Associated with Generative-AI" has now been seen by Reviewer #2, whose comments appear below. In light of their advice, I am delighted to say that we are happy, in principle, to publish it in *Communications Psychology* under a Creative Commons 'CC BY' open access license.

We will not send your revised paper for further review if the editorial requests and the referees' comments on the present version have been addressed.

I have attached an edited version of your manuscript, and ask you to attend to each comment in detail.

Phrased as a high-level request, we ask that you revise the manuscript to refrain from unnecessary speculation to highlight the dangers of Gen AI (see ref report: it's possible, but not clear whether having access to psychological research about trust and persuasion will allow generation of more persuasive, seemingly trustworthy messaging/manipulation). Instead, the piece should focus on the contribution that psychology/psychologists need to make in the realm of Gen AI, as indicated in the preface. Any illustrative examples should be chosen so that they tie in with the general theme of the work, rather than distract from the central messaging. The target audience is researchers in psychology.

EDITORIAL REQUESTS:

* Please review the changes in the attached copy of your manuscript, which has been edited for style, and address the comments and queries I have added. If using Word, please use the 'track changes' feature to make the process of accepting your manuscript more efficient.

* Please check whether your manuscript contains third-party images, such as figures from the literature, stock photos, clip art or commercial satellite and map data. If any of the display items in your manuscript (figures, tables, boxes or movies) include images that are the same as, or are adaptations of, previously published images, please fill in the Third Party Rights Table, and return to us when you submit your revised manuscript. This information will enable us to obtain the necessary rights to re-use such material. If we are unable to obtain the necessary rights to use or adapt any of the material that you wish to use, we will contact you to discuss alternative options.

* Communications Psychology uses a transparent peer review system. On author request, confidential information and data can be removed from the published reviewer reports and rebuttal letters prior to publication. If you are concerned about the release of confidential data, please let us know specifically what information you would like to have removed. Please note that we cannot incorporate redactions for any other reasons.

*If you have not done so already, please alert me to any related manuscripts from your group that are under consideration or in press at other journals, or are being written up for submission to other journals (see www.nature.com/authors/editorial_policies/duplicate.html for details).

FORMATTING GUIDELINES:

You will find a complete list of formatting requirements following this link:
<https://www.nature.com/documents/commsj-style-formatting-checklist-comment.pdf>

Please use the checklist to prepare your manuscript for final submission. In the following, I also highlight some issues of particular importance.

** Main text

Please provide three or four section headings in the main text. These should relate to the content of the article rather than being generic. Headings should be no longer than 30 characters (including spaces) and should not use punctuation.

** Figures

Please remove all figures from the main text and upload them individually, one figure per file. To ensure the swift processing of your paper please provide the highest quality, vector format, versions of your images (.ai, .eps, .psd) where available. Text and labelling should be in a separate layer to enable editing during the production process. If vector files are not available then please supply the figures in whichever format they were compiled in and not saved as flat .jpeg or .TIFF files. If your artwork contains any photographic images, please ensure these are at least 300 dpi.

* Figures should be simple and informative — multi-part figures are best avoided.

* References

The upper limit for References is 10. References appear as superscript Arabic numerals, in order of mention. The reference list mentions references in the numerical order in which they are mentioned in the main text. If a reference is cited more than once, the same number is used throughout the text and the reference receives a single entry in the reference list.

Only papers that have been published or accepted by a named publication should be in the reference list (preprints and citations of datasets are also permitted). Unpublished/Submitted research should not be included in the reference list; it should only be mentioned briefly and parenthetically in the main text. Note that no major arguments should rely on unpublished research.

Published conference abstracts and URLs for websites should be cited parenthetically in the text, not in the reference list.

Footnotes are not used.

* Competing interests

Please include a "Competing interests" statement after the References. Note that we ask authors to declare both financial and non-financial competing interests. For more details, see <https://www.nature.com/authors/policies/competing.html>. If you have no financial or non-financial competing interests, please state so: "The authors declare no competing interests."

SUBMISSION INFORMATION:

In order to accept your paper, we require the following:

* A cover letter describing your response to our editorial requests.

* A separate document detailing your point-by-point response to any issues raised by our referees (please include the referees' comments in this document).

* The final version of your text as a Word or TeX/LaTeX file, with any tables prepared using the Table menu in Word or the table environment in TeX/LaTeX and using the 'track changes' feature in Word.

* Production-quality versions of all figures, supplied as separate files. Photographic images should be 300 dpi in RGB format (.jpg, TIFF or native Photoshop format) and any labels/scale bars included in a separate layer from the image. Line art, graphs and schemes should be vector format (.ai, .eps, .pdf); Adobe Illustrator files are preferred and will minimize production time. Any chemical structures or schemes contained within figures should additionally be supplied as separate Chemdraw (.cdx) files.

At acceptance, the corresponding author will be required to complete an Open Access Licence to Publish on behalf of all authors, declare that all required third-party permissions have been obtained.

Please note that your paper cannot be sent for typesetting to our production team until we have received this information; therefore, please ensure that you have this ready when submitting the final version of your manuscript.

ORCID

Communications Psychology is committed to improving transparency in authorship. As part of our efforts in this direction, we are now requesting that all authors identified as 'corresponding author' create and link their Open Researcher and Contributor Identifier (ORCID) with their account on the Manuscript Tracking System (MTS) prior to acceptance. ORCID helps the scientific community achieve unambiguous attribution of all scholarly contributions. For more information please visit <http://www.springernature.com/orcid>

For all corresponding authors listed on the manuscript, please follow the instructions in the link below to link your ORCID to your account on our MTS before submitting the final version of the manuscript. If you do not yet have an ORCID you will be able to create one in minutes.

IMPORTANT: All authors identified as 'corresponding author' on the manuscript must follow these instructions. Non-corresponding authors do not have to link their ORCIDs but are encouraged to do so. Please note that it will not be possible to add/modify ORCIDs at proof. Thus, if they wish to have their ORCID added to the paper they must also follow the above procedure prior to acceptance.

To support ORCID's aims, we only allow a single ORCID identifier to be attached to one account. If you have any issues attaching an ORCID identifier to your MTS account, please contact the Platform Support Helpdesk.

[link redacted]

We hope to hear from you within two weeks; please let us know if the process may take longer.

Best regards,

Marike

Marike Schiffer, PhD

Chief Editor

Communications Psychology

REVIEWERS' COMMENTS:

Reviewer #2 (Remarks to the Author):

This is a revised version of a comment. I think the authors made progress but I still have two fundamental concerns about this work.

First, the authors address previous concerns by highlighting that the use of generative AI can lead to adverse outcomes such as influencing democratic political processes and even terrorism. While this might be conceivable, readers might not understand the role of generative AI in this context. In other words, the authors warn that the use of gen AI can lead to such outcomes but without more detailed insights and examples of how this would be implemented, this claim might be less credible and highly speculative. It would also be important to clarify how content produced by generative AI can be different from what we already observe in the real world (e.g., we already have figures who spread misinformation on a large scale; we also have actors who know well how to use persuasion principles). In sum, my impression is that the current write-up is too abstract and would only unfold its potential if it contained more concrete details.

Second, the manuscript implies that psychological findings could pose dangers similar to those in disciplines regulated by Export Control legislation, like chemical weapons development. This comparison might seem exaggerated to many readers. Historical examples and literature, including biographies of influential figures, suggest that manipulative skills are not solely derived from psychological research. While influential works like Cialdini's book on persuasion could be misused, it is questionable whether typical psychological studies hold similar risks. The paper seems to paint an overly grim view of psychology's potential risks. I would welcome evidence to the contrary from the authors.

On a minor note, the subheading "Why should Psychologists be (more) concerned?" seems misplaced. The content under this heading discusses the implications for the field of psychology rather than for psychologists specifically.

Response to Reviewer

COMMSPSYCHOL-23-0438A, “How and Why Psychologists Should Respond to the Harms Associated with Generative-AI”

Below, we outline the reviewer’s suggestions (indented, in italics) and how we have addressed them in this revision (in bold).

Reviewer #2

- 1. First, the authors address previous concerns by highlighting that the use of generative AI can lead to adverse outcomes such as influencing democratic political processes and even terrorism. While this might be conceivable, readers might not understand the role of generative AI in this context. In other words, the authors warn that the use of gen AI can lead to such outcomes but without more detailed insights and examples of how this would be implemented, this claim might be less credible and highly speculative. It would also be important to clarify how content produced by generative AI can be different from what we already observe in the real world (e.g., we already have figures who spread misinformation on a large scale; we also have actors who know well how to use persuasion principles). In sum, my impression is that the current write-up is too abstract and would only unfold its potential if it contained more concrete details.*

We have made three types of revisions to make this version more concrete than the previous version, whilst keeping within the word and reference limits:

- (1) We have removed any speculative content (as per the Editor’s request), provide a reference to evidence each known adverse outcome, and we highlight that anticipating and understanding mechanisms of influence is a priority direction for future research (this is the motivations for this *Comment*), e.g.,**

“A psychological research agenda should prioritize research that aims to understand how the features and affordances of Gen-AI technology interact with social and psychological variables. For example, research could investigate the factors that affect people’s vulnerability and resilience to being influenced by AI-generated media, through investigating how psychological variables moderate the impact of the affordances of specific technical features on (for example) the perceived trustworthiness, persuasiveness, humanness, and relatability of such media – and outcomes such as attitude, norm, and behavior change.” (pp. 6-7)

- (2) We provide examples, e.g.,**

“The move towards open science means that our insights published in open access papers could be available as training data for LLMs. That is, Gen-AI tools like ChatGPT can synthesize the outputs of psychological research and use this to suggest more advanced ways to design synthetic media to manipulate and influence a target audience. Insights from Psychological research are known to have been used to

influence and manipulate people: for example, data on personality and social media behavior were used by Cambridge Analytica to influence election outcomes.” (p. 5)

- (3) To help psychologists to anticipate and consider potential future outcomes of technological innovations, we then point to the responsible innovation framework (which describes a series of concrete methodological steps), and provide an example of how it has been used, e.g., pp. 7-8:

“Third, to better anticipate (and prevent) harms by design, psychologists need to collaborate with technologists and academics from other disciplines using mechanisms such as the responsible research innovation framework of UKRI. This would generate questions that would not otherwise be asked, leading to a better understanding of the assumptions baked into Gen-AI and the underlying ML that would not otherwise be recognized. This interdisciplinary approach has already been used, for example, to identify algorithmic contingencies that can affect future outcomes, uncertainties, and harms of ML models¹⁰. In the same way, psychologists can help identify contingencies that could affect the potential of Gen-AI technologies to cause - or prevent - harm. In particular, the “Anticipate - Reflect - Engage - Act” (AREA) model¹ can help psychologists to generate research questions and elucidate the factors that can give rise to contextual vulnerability.”

In doing so, we aim to help equip psychologists with a research agenda and a methodological tool that will enable them to leverage their own theories to predict and explain how the design of technology relates to potential outcomes.

2. *Second, the manuscript implies that psychological findings could pose dangers similar to those in disciplines regulated by Export Control legislation, like chemical weapons development. This comparison might seem exaggerated to many readers. Historical examples and literature, including biographies of influential figures, suggest that manipulative skills are not solely derived from psychological research. While influential works like Cialdini’s book on persuasion could be misused, it is questionable whether typical psychological studies hold similar risks. The paper seems to paint an overly grim view of psychology’s potential risks. I would welcome evidence to the contrary from the authors.*

We did not mean to imply that all (or “typical”) psychological research equates to research on chemical weapons. We have now removed the sentences referring to export control and references to weaponization of insights, and in our previous revision we had added a sentence referring to example benefits of AI, so as to avoid painting an overly grim view (see our response letter to our original submission, response to Reviewer 1 # 4).

We also note on p. 4 that:

“it is not only these nefarious uses that should be of concern to psychologists: it is also the legitimate use of such hyper realistic synthetic media, from business and marketing to education and entertainment.”

Benefits of Gen-AI are described as follows:

“This has benefits, for example, Gen-AI can potentially improve the efficiency of creative work.” (p. 3)

And we describe a hopeful outlook:

“Furthermore, by using the output of Gen-AI to understand biases that are embedded in training data, and how the outputs affect people, psychologists can advise on how LLMs can use more appropriate training data, and even curate or synthesize training datasets that may work with Gen-AI to reduce societal inequalities. This means that psychologists can help others to use Gen-AI for the betterment of - rather than to the detriment of - society. Through research in these areas, raising our collective voice, and engaging responsibly with government, regulators, industry, and other disciplines, Psychologists can help ensure a future in which Gen-AI works to support - rather than harm – people and society.” (p. 8)

- 3. On a minor note, the subheading "Why should Psychologists be (more) concerned?" seems misplaced. The content under this heading discusses the implications for the field of psychology rather than for psychologists specifically.*

We have re-titled the paper, “How and Why Psychologists Should Respond to the Harms Associated with Generative-AI”. We have checked the manuscript for our usage of the term “psychology” versus “psychologists” and have made edits to ensure we use the latter term. We have endeavoured to reflect what psychologists (rather than psychology) can do in the section entitled, “A Research Agenda”.